# Abstractive Open Information Extraction

**Kevin Pei**[1], **Ishan Jindal**[2], **Kevin Chen-Chuan Chang**[1]
[1]University of Illinois at Urbana-Champaign, [2]IBM Research
[1]{kspei2, kcchang}@illinois.edu, [2]ishan.jindal@ibm.com

## Abstract

Open Information Extraction (OpenIE) is a traditional NLP task that extracts structured information from unstructured text to be used for other downstream applications.Traditionally, OpenIE focuses on extracting the surface forms of relations as they appear in the raw text, which we term extractive OpenIE. One of the main drawbacks of this approach is that implicit semantic relations (inferred relations) can not be extracted, compromising the performance of downstream applicationsIn this paper, we broaden the scope of OpenIE relations from merely the surface form of relations to include inferred relations, which we term **abstractive OpenIE**. This new task calls for the development of a new abstractive OpenIE training dataset and a baseline neural model that can extract those inferred relations. We also demonstrate the necessity for a new semantics-based metric for evaluating abstractive OpenIE extractions. Via a case study on Complex QA, we demonstrate the effectiveness of abstractive OpenIE. [1]

## 1   Introduction

Open Information Extraction (OpenIE) is the task of extracting relation tuples from unstructured text (Etzioni et al., 2008; Mausam et al., 2012; Angeli et al., 2015). Unlike traditional information extraction, OpenIE is open domain, intended to be easy to deploy in different domains without fine-tuning. These relations can then be used in downstream applications like summarization (Zhang et al., 2021), question-answering (Lu et al., 2019), and knowledge base population (Kroll et al., 2021). In order to support these applications, OpenIE needs to extract as many different types of relations as possible. One particular relation type of interest is "Inferred Relations". We define an "Inferred

---

[1]Code and models are available at https://github.com/kevinpei/AbstractiveOpenIE

| Sample Sentence | Tokyo, officially Tokyo Metropolis, is the capital city of Japan and one of its 47 prefectures. |
|---|---|
| Extractive OpenIE Extractions | {Tokyo; is; the capital city of Japan} {Tokyo; is; one of its 47 prefectures} |
| Abstractive OpenIE Extractions | {Tokyo; is; the capital city of Japan} {Tokyo; is officially; Tokyo Metropolis} {Tokyo; is; a prefecture} or {Tokyo; is; one of Japan's 47 prefectures} |

Table 1: Examples of relations that extractive OpenIE models can not extract. In this sentence, the apposition "officially Tokyo Metropolis" has no predicate but still has a relation with the noun "Tokyo". In the last abstractive relation, "one of its 47 prefectures" is meaningless without the context of the rest of the sentence. It would be more useful to replace the object with "a prefecture" or "one of Japan's 47 prefectures", neither of which appear in the sentence. Preexisting OpenIE models can not extract these abstractive relations.

Relation" to be a relation where the predicate contains words that are not in the original sentence. For example, given the sentence "*Albert Einstein (14 March 1879 - 18 April 1955) was a German-born theoretical physicist*", the relation *(Albert Einstein, died on, 18 April 1955)* can be inferred even though "died on" is not in the original sentence. Extracting inferred relations increases recall, which is explicitly desired by various downstream tasks including question-answering, slot filling, event schema induction, summarization, and knowledge base population (Pei et al., 2022). Based on the number of inferred relations in the manually annotated dataset WiRe57, extracting inferred relations could increase the total number of relations extracted by 50% (Léchelle et al., 2018). Existing neural OpenIE models struggle to extract these inferred relations, with only one previous model, OpenIE6, including hand-written rules to extract only some cases of inferred relations (Kolluru et al., 2020a). Table 1 has an example of an inferred relation.

Another problem is that the extraction is very dependent on the sentence's syntax. For downstream

applications using OpenIE, it is important to be able to extract either different surface forms of a relation or its canonical form. The surface form refers to how it appears within the text, while the canonical form refers to the semantic meaning. In question answering (QA), several methods repeatedly paraphrase the questions so that the surface forms of extracted relations match at least one of the question paraphrases, indicating that extracting more surface forms of relation would answer more questions (Fader et al., 2013, 2014; Yin et al., 2015). In addition, the more complex a sentence's syntax is, such as having more clauses, the more difficult it is to extract high-quality relations. An illustrative example of how being limited to extracting surface forms can be found in Table 1.

By design, all existing neural OpenIE models are unable to extract these abstractive relations, which could be utilized by the downstream application. Therefore, in this work, we propose an abstractive Open Information Extraction (abstractive OpenIE) task. The purpose of this task is to extract relation tuples that are far beyond the reach of any existing OpenIE tasks. We define abstractive OpenIE as a task that given an input sentence generates ordered tuples in the form of (subject, predicate, object) for all possible relations (inferred or non-inferred) within the sentence.

Although not explicitly defined as such, existing neural models often treat OpenIE as a labeling problem, where tokens are labeled as being part of the subject, predicate, or object of a relation (Kolluru et al., 2020a; Vasilkovsky et al., 2022). Even in cases where OpenIE is defined as a generative problem, the generated relations don't contain words outside the vocabulary of the original sentence (Kolluru et al., 2020b) (Han and Wang, 2021). Due to the labeling problem definition, prior neural OpenIE models struggle to extract relations with predicates that don't appear in the original sentence. We refer to all preexisting neural OpenIE models as **extractive OpenIE** methods, because they can only generate relations by extracting tokens from the original sentence.

One such attempt to go beyond extractive OpenIE is the OpenIE6 model Kolluru et al. (2020a). It explicitly concatenates manually defined out-of-vocabulary tokens at the end of each sentence to allow for the extraction of specific inferred relations. However, obtaining such a list is non-trivial and can not scale to every domain. We differ from OpenIE6 in the sense that abstractive OpenIE models trained on abstractive OpenIE training datasets generate this inferred relation on the fly and do not require defining a list of out-of-vocabulary tokens. Therefore, in this paper, we derive abstractive OpenIE training datasets from existing information extraction datasets and train a baseline machine-learning model that extracts abstractive relations.

Further, we also develop an abstractive OpenIE evaluation metric to evaluate the quality of abstractive OpenIE models. Our problem warrants a new evaluation metric because all the existing OpenIE evaluation metrics are lexical and evaluated based on the token overlap between the predicted relations and the gold standard relation. These lexical metrics are undesirable for the proposed task as the relations extracted using the abstractive OpenIE model do not have to use the tokens present in the input sentence. Therefore, we propose a semantics-based metric for evaluating abstractive OpenIE models.

In summary, our contributions are as follows:

- We propose an **abstractive OpenIE** task to expand the scope of OpenIE extractions compared to prior extractive OpenIE models.

- We derive an abstractive OpenIE training dataset and develop an initial abstractive OpenIE model as a baseline.

- We propose a general-purpose semantics-based evaluation metric for evaluating any OpenIE model.

- We perform a comprehensive comparison between abstractive and extractive OpenIE models.

## 2   Related Work

**OpenIE Datasets:** Given how data-hungry deep learning models are and how costly it is to manually label OpenIE datasets, most OpenIE training sets are weakly labeled using high-confidence extractions from prior OpenIE models to get "silver-standard" labels. For example, the CopyAttention (Cui et al., 2018), SpanOIE (Zhan and Zhao, 2020), and OIE4 (Kolluru et al., 2020b) training sets are created from high-confidence OpenIE4 extractions from Wikipedia. LSOIE (Solawetz and Larson, 2021) is instead created from examples from the QA-SRL 2.0 dataset. Because traditional OpenIE is

| | Dataset | Number of Sentences | Number of Relations | Number of Relations with Inferred Predicates | Number of Relations with Inferred Predicates or Arguments |
|---|---|---|---|---|---|
| Training Sets | OIE4 | 90K | 160K | 0 | 0 |
| | OIE4 Backtranslated | 44K | 61K | 19K | 48K |
| | OIE4 with SuRE Relations | 90K | 178K | 16K | 16K |
| | OIE4 Backtranslated with SuRE Relations | 44K | 69K | 26K | 56K |
| Test Sets | WiRe57 | 57 | 343 | 116 | 120 |
| | CaRB | 634 | 2715 | 736 | 798 |
| | ReOIE2016 | 683 | 1508 | 155 | 156 |
| | LSOIE | 2402 | 5371 | 0 | 0 |

Table 2: Comparison of the attributes of different datasets. SuRE is the relation extraction model we use to obtain additional inferred relations for training (Lu et al., 2022).

| | |
|---|---|
| Sample Sentence | The purse contains the seal of Order of the Garter. |
| Back Translated Sentence | In the handbag is the seal of the Order of the Garter. |
| Relations | {The purse; contains; the seal of Order of the Garter} |

Table 3: An example of paraphrasing via back translation. The sentence is from the OIE4 training set.

extractive, there are no inferred relations in OpenIE training sets, with only hand-labeled benchmarks containing inferred relations. As a result, these training sets are not well-suited for training an abstractive OpenIE model.

In contrast, there are several benchmarks with inferred relations. WiRe57 (Léchelle et al., 2018) is 57 manually annotated sentences. CaRB (Bhardwaj et al., 2019) uses crowdsourcing to re-annotate the sentences in the OIE2016 benchmark, the first commonly used OpenIE benchmark (Stanovsky and Dagan, 2016). ReOIE2016 (Zhan and Zhao, 2020) is a different manual re-annotation of OIE2016 to attempt to resolve problems arising from incorrect extractions. LSOIE also has its own benchmark created using the same method as its training set. WiRe57, CaRB, and ReOIE2016 all contain inferred relations, making them useful for evaluating abstractive OpenIE.

**OpenIE Models:** OpenIE6 is a neural OpenIE model that performs BIOES tagging for the subject, predicate, and object of each relation (Kolluru et al., 2020a). At the end of each sentence, it appends the tokens "be", "of", and "from" so that they can also be tagged as part of the predicate. However, this method limits inferred relation extraction to only those containing the tokens they manually specify and doesn't help with the issue of extracting only the surface form of the relation.

IMoJIE is an OpenIE model that tries to reduce the redundancy of relations by appending extracted relations to the end of each sentence (Kolluru et al., 2020b). This new sentence is then given as input so the model can identify what relations have previously been extracted at the cost of significantly reduced extraction speed. Although it uses a generative neural model, IMoJIE relies on its copy mechanism to extract relations, so its vocabulary is limited so that it only generates tokens that are within the original sentence. In addition, the focus on reducing redundancy means it is also constrained to extracting only a single surface form of each relation in each sentence.

Gen2OIE is an OpenIE model that fine-tunes a seq2seq model to generate relations (Kolluru et al., 2022). It follows a two-stage approach, where predicates are first extracted, then arguments are extracted for each predicate. Unlike previous OpenIE models, Gen2OIE can generate relations using tokens that do not appear in the original sentence.

Closed Information Extraction (CIE) is a related task where relations within an existing KB are extracted from unstructured text. GenIE proposes a generative model to perform this task (Josifoski et al., 2021). However, CIE is an inherently more limited task than OpenIE due to its dependence on a preexisting domain. CIE models are unable to extract relations from new and emerging domains and require human effort to transfer to new domains.

**OpenIE Evaluation Metrics:** Existing OpenIE metrics are lexical. This means that extracted relations are evaluated based on the token overlap

| Sample Sentence | In 569, unopposed, Alboin took northern Italy's main city, Milan. |
|---|---|
| Extractive Relations | {Alboin; took; In 569 nothern Italy's main city, Milan} |
| SuRE-Extracted Relations | {northern Italy's main city; is also known as; Milan} |

Table 4: An example of data augmentation via relation extraction. The method used for relation extraction is SuRE (Lu et al., 2022). The sentence is from the OIE4 training set.

between the predicted relations and the gold standard relations. In particular, OIE2016 is based on tuple-level matching, treating relations extraction as a binary classification problem where a gold standard relation is extracted if a predicted relation contains a majority of tokens in the gold standard relation (Stanovsky and Dagan, 2016). WiRE57 and CaRB use token-level matching, where predicted relations are evaluated based on the token overlap between the best matches between the predicted and gold standard relations (Léchelle et al., 2018) (Bhardwaj et al., 2019). Because the abstractive relations extracted using abstractive OpenIE do not have to use the original sentence's tokens, evaluating them using lexical metrics is undesirable.

There has been previous interest in semantics-based metrics for evaluating abstractive summarization and machine translation. BERTScore is a popular metric that calculates the cosine similarity between the BERT contextual embeddings of each token in the document and each token in the summary. The highest total similarity score possible from the mapping of tokens in the document to tokens in the summary is then chosen as the BERTScore (Zhang et al., 2019). In theory, this metric would take into account the context of each word, which would capture the semantics of each word. However, it has been found that BERTScore may still be insufficient in cases where individual tokens like negations significantly change the meaning of the sentence, even if it is marginally better than lexical methods like BLEU, ROUGE, and METEOR (Saadany and Orasan, 2021).

## 3  Abstractive OpenIE

Abstractive OpenIE is defined as a task that generates ordered tuples in the form of (subject, predicate, object) for all possible relations (inferred or non-inferred) within a given sentence. In this section, we will describe all the pieces required to accomplish this task.

### 3.1  Training Sets

Although there are existing OpenIE training sets, they do not fit our goals because they are purely extractive. The training set needs to contain inferred relations so that trained models can extract inferred relations. To address this problem, we use two methods to derive abstractive OpenIE training sets from OIE4, a preexisting OpenIE training set:

**Paraphrasing Via Back Translation**

Back translation is the translation of a text into a different language, then translation back into the original language (Edunov et al., 2018). The resulting text should retain the same semantic meaning, but may differ in the specific words or syntax used. To generate abstractive OpenIE training data, we generate back translations of the sentences but retain the gold standard relations. Because the back translated sentences use different words and syntax, the gold standard relations may no longer consist of only words from the original sentence, thus becoming inferred relations. We provide an example in Table 3.

When generating paraphrases, we need to make sure that the paraphrased sentence has the same semantic meaning as the original sentence and contains the same relations. Thus, we perform a validation step where we use entailment to measure the quality of the paraphrase. During this step, we use three measures to ensure the quality of the paraphrase. We measure whether the original sentence entails the paraphrase to ensure the paraphrase doesn't contain extraneous information not in the original sentence. We measure whether the paraphrase entails the original sentence to ensure the paraphrase contains all information present in the original sentence. Finally, we measure whether the paraphrased sentence entails all of the gold standard relations to ensure that the relations are the same for the original sentence and the paraphrase. If any of these hypotheses does not have an entailment confidence above a certain threshold, then we do not use the paraphrase in the training data.

**Data Augmentation Via Relation Extraction**

Although paraphrasing can create inferred rela-

| Sample Sentence | Sharon had been in a coma since suffering a stroke in January 2006. |
|---|---|
| Sample Relations | {Sharon; had been; in a coma}
{Sharon; suffering; a stroke in January 2006} |
| Sample Predicate
Prediction Input | predicates: Sharon had been in a coma since suffering a stroke in January 2006. [pred] had been [pred] suffering |
| Sample Argument
Prediction Inputs | args: Sharon had been in a coma since suffering a stroke in January 2006. [pred] had been [arg1] Sharon [arg2] in a coma
args: Sharon had been in a coma since suffering a stroke in January 2006. [pred] had been [pred] suffering [arg1] Sharon [arg2] a stroke in January 2006 |

Table 5: Illustrative training example. For each sentence, there is one predicate prediction example and a number of argument prediction examples equal to the number of gold standard relations. The model first extracts all predicates, then for each predicate extracts the arguments.

tions in that the words used may not match the sentence exactly, the relations remain fundamentally the same. The inferred relations that the benchmarks such as WiRe57 contain are not derived from paraphrases of the sentence, so creating paraphrases as training data for them is not appropriate. Instead, we augment the data with additional inferred relations derived using relation extraction (RE). We provide an example in Table 4.

RE also aims to extract relations from unstructured text, but instead of being completely open domain, RE is limited to extracting a specific set of relations that must be defined beforehand (Bach and Badaskar, 2007). However, those relations may take a variety of surface forms. For instance, the relation "country_of_birth" could take the form "Einstein was born in Ulm", "Einstein (born 14 March 1879 in Ulm)", other forms. We thus use RE models to extract additional inferred relations for abstractive OpenIE training. To ensure quality and prevent redundancy, we only keep extracted relations above a certain level of confidence and which are not entailed by or entail preexisting OpenIE gold standard relations.

## 3.2 Benchmarks

In contrast to existing OpenIE training dataset, there are several OpenIE benchmarks which contain inferred relations because they were manually annotated or used crowdsourcing for annotation. For evaluation, we use WiRe57, CaRB, Re-OIE2016, and LSOIE test sets. Each of these benchmarks contains a different proportion of inferred relations, in Table 2. In particular, the manual annotation of WiRe57 makes prior extractive OpenIE methods perform poorly compared to their performance on other OpenIE benchmarks. Unlike the other benchmarks, LSOIE contains no inferred relations at all, meaning in theory extractive OpenIE methods should be able to extract all relations. Thus, we can use performance on LSOIE to directly compare abstractive OpenIE and extractive OpenIE models on the extractive OpenIE task.

Statistics for the derived training sets and benchmarks is available in Table 2.

## 3.3 Abstractive Tuple Generator

Prior OpenIE models are not suited for the proposed task because all existing models are extractive models. As a result, we use generative models to generate relations for a given sentence. We choose to fine-tune T5, a text-to-text transformer model, to generate relations from a sentence (Raffel et al., 2020).

Inspired by Multi[2]OIE, we perform relation generation in two stages, a predicate and an argument stage (Ro et al., 2020). In the predicate stage, all predicates are extracted from the sentence at once. The input for this stage is the sentence, while the gold standard is the predicates of all gold standard relations separated by the special "[pred]" token. Although the order of relations in our output doesn't matter, we need to enforce a strict order for the model to learn. Thus, we order the predicates by their position within the sentence.

For the argument prediction stage, for each predicate the model predicts the arguments for the relation with that predicate. Because multiple relations may have the same predicate, we specify the predicate by including all predicates before it in the sentence. For each relation, we assume there are two arguments, which the model extracts simultaneously. The input for this stage is the sentence with the predicate concatenated to the end, separated by

a "[pred]" special token, while the gold standard is the arguments for the gold relation corresponding to that predicate, in Table 5.

## 3.4 Semantic-based Evaluation Metrics

CaRB is a popular metric for evaluating OpenIE models, but it requires the predicates of the prediction and gold standard to match to score a given prediction. Although it serves as a good proxy for a semantic metric in extractive OpenIE, it is significantly less useful for abstractive OpenIE where the space of all possible predicates is much larger than just the tokens in the sentence.

To evaluate abstractive OpenIE, we require a semantics-based metric rather than a lexical metric based on token matching. Although previous semantics-based evaluation metrics like BERTScore exist, we do not find them to be appropriate for our use case. Previous semantics-based evaluation metrics do not work well for cases where a single token can dramatically change the semantics of a statement, such as negations like "not" (Saadany and Orasan, 2021). Thus, we introduce a set of 3 evaluation metrics based on entailment for more accurate semantic evaluation. Each of these metrics measures semantic coherence at different granularities, and which granularity is most important will depend on the application and properties of the datasets. We demonstrate this necessity with an example in Table 6.

When calculating the entailment score for a relation, we remove special characters so that it resembles a sentence. For instance, for the relation triple {Sharon; had been; in a coma}, we form the statement "Sharon had been in a coma."

**Sentence-tuple entailment** The first metric we propose is sentence-tuple entailment. For recall, we combine all the relations together and see if the combined relations entail the sentence. If the combined relations do not entail the sentence, that means the sentence contains information not in any relation and thus the extracted relations as a whole have poor recall. For precision, we take the average of the entailment score obtained when seeing if the sentence entails an individual relation for all extracted relations. If the relation is not entailed, that means it contains information not in the sentence and thus has poor precision.

**Combined tuple-tuple entailment** The second metric we propose is combined tuple-tuple entailment. This metric is inspired by a metric proposed

by (Dušek and Kasner, 2020). For this metric, we use the gold standard relations to evaluate the extracted tuples. The combined tuple in this case refers to the combination of all gold standard relations. For recall, we combine all the predicted relations together and see if the combined relations entail the combined gold relations. If the combined predictions do not entail the combined gold, that means the gold relations contains information not in any prediction and thus the extracted relations as a whole have poor recall. For precision, we take the average of the entailment score obtained when seeing if the combined gold entails an individual relation for all extracted relations. If the prediction is not entailed, that means it contains information not in any gold relation and thus has poor precision. Compared to the sentence-tuple entailment metric, this one excludes any extraneous information in the sentence not in the gold standard relations from evaluation.

**Tuple-tuple entailment** The third metric we propose is tuple-tuple entailment. This metric is based on the OpenIE metric CaRB (Bhardwaj et al., 2019). For recall, for each gold standard relation we calculate the entailment for each extracted relation if the gold standard entails that prediction. Then, for each gold standard relation its recall is equal to the highest entailment score achieved by any of the predictions. The recall for the sentence is the average of the recall of its relations. Note that the highest recall for multiple gold standard relations can be achieved by the same predicted relation if the predicted relation contains all of those gold standard relations. For precision, for each gold standard relation we calculate the entailment for each extracted relation if the prediction entails that gold standard relation. Then, we find the optimal matching of gold standard relations to extracted relations that results in the highest average precision. Unlike recall, when calculating precision a predicted relation can only entail a single gold standard relation. This is because we want the number of predictions to match the number of gold relations.

## 4 Experimental Setup

**Datasets and Metrics**
We evaluate the trained abstractive OpenIE model on four benchmarks: WiRe57, CaRB, Re-OIE2016, and LSOIE-wiki with respectively decreasing proportion of inferred relations.

| Sentence | Rival political factions were unable to resolve disagreements. | | | |
|---|---|---|---|---|
| Gold Standard | {Rival political factions unable to; resolve; disagreements} | | | |
| Prediction | {Rival political factions to; resolve; disagreements} | | | |
| Evaluation Metric | CaRB F1 Score | ROUGE-1 Score | BERTScore F1 Score | Tuple-Tuple Entailment F1 Score |
| | 0.923 | 0.923 | 0.976 | 0.005 |

Table 6: Comparison of different evaluation metrics on an example from the training set. CaRB is a popular lexical metric used to evaluate OpenIE (Bhardwaj et al., 2019). ROUGE-1 is a popular lexical metric to evaluate summarization (Lin, 2004). BERTScore is a previous semantics-based metric used to evaluate summarization (Zhang et al., 2019). Tuple-Tuple Entailment is a new semantics-based metric we propose.

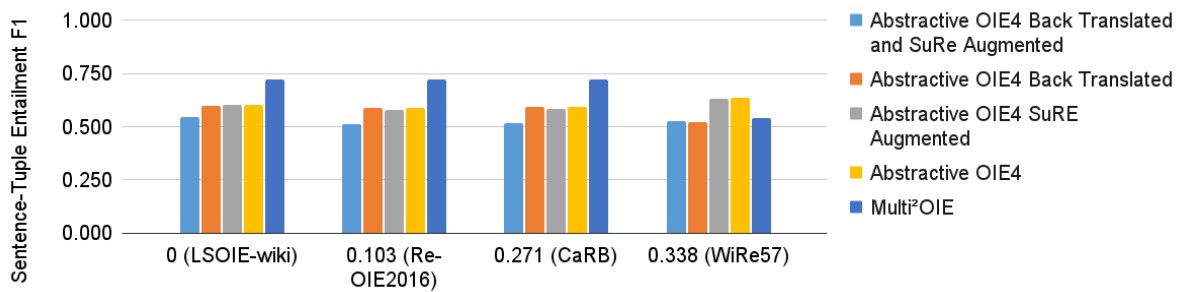

Figure 1: Comparison of Sentence-Tuple Entailment F1 Score of different OpenIE models on all relations in the benchmarks. All models are trained on OIE4.

Since OIE4 trained OpenIE models showed superior F1 performance on all these benchmarks as compared to other OpenIE training sets we derive abstractive training data from this dataset. We generate four different versions of OIE4 using the methods we describe in Section 3.1. The first version is the original extractive dataset, the second version uses backtranslation for paraphrasing, the third version is augmented by relation extraction, and the fourth uses both backtranslation and relation extraction for augmentation. For backtranslation we use Facebook-FAIR's WMT'19 German-English and English-German models (Ng et al., 2019) and retain only those back translated sentences whose entailment confidence is above 80%. For relation extraction, we use a pretrained SuRE model, a state-of-the-art relation extraction model (Lu et al., 2022) without any additional fine-tuning and keep all relations with confidence above 80%. These confidence thresholds are hyperparameters that may be adjusted.

We compare performance using the preexisting CaRB metric, as well as our own introduced semantics-based metrics of tuple-tuple entailment,

combined tuple-tuple entailment, and sentence-tuple entailment. The entailment model we use for our datasets and evaluation metrics is a BERT-based encoder model trained on MNLI, SNLI, and the Hans dataset (Gao et al., 2021).

**Models and Hyperparameters**

We fine-tune the T5-base model for our experiments. We fine-tuned T5 for 5 epochs with an initial learning rate of 2e-5 and batch size of 12. We validate T5 on a subset of the OIE4 training set using the tuple-tuple entailment metric. We also compare our model with Multi[2]OIE, a state-of-the-art neural extractive OpenIE model (Ro et al., 2020). We train Multi[2]OIE on the original OIE4 dataset with no paraphrasing. We use the default hyperparameters of Multi[2]OIE.

## 5 Results and Analysis

For this section, we focus on the sentence-tuple semantic score because it offers a holistic comparison of the extracted relations and the sentence and does not rely upon potentially incomplete or faulty gold relations. Full tables with our empirical results in-

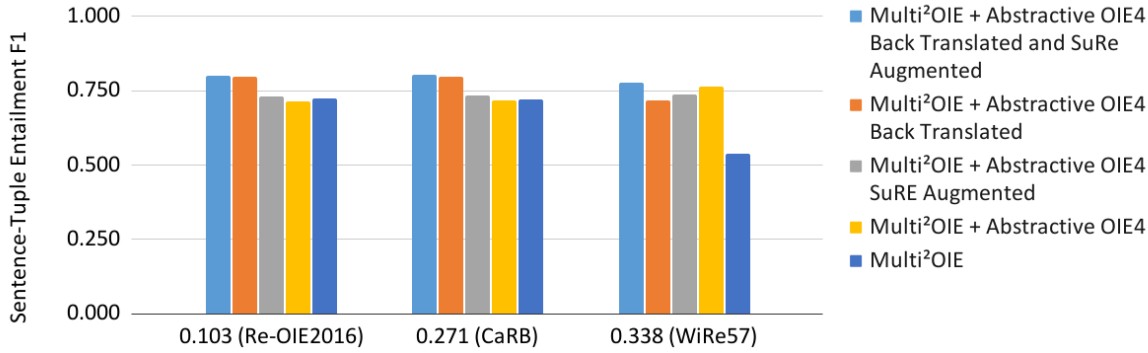

Figure 2: Comparison of Sentence-Tuple Entailment Recall of different combinations of OpenIE models on all relations in the benchmarks. All models are trained on OIE4.

cluding other metrics can be found in Appendix A.

We first compare performance on all relations in Figure 1. In general, abstractive OpenIE leads to better performance the higher the proportion of inferred relations in the test set. This is expected because Multi[2]OIE can not extract inferred relations at all. When considering the full benchmarks, of the data augmentation methods we use, SuRE augmentation works the best. Training on back-translated OIE4 degrades the performance compared to the base extractive OIE4 data. This may be because back translation reduces the amount of training data. Additionally, back translation often just replaces the gold standard predicate with a synonym instead of changing the syntax of the sentence, which does not help in the extraction of inferred relations.

To demonstrate the complementary nature of abstractive OpenIE to extractive OpenIE, we combine their extractions. When combining their extractions, we remove redundant relations by removing relations that are entailed by any other relations. If two relations entail each other, then we keep the longer one. A comparison of combined models can be found in Figure 2. When combining model predictions, we observe that back translation actually helps more than SuRE augmentation. This suggests that SuRE augmentation helps extractive OpenIE relations, while back translation is more useful for increasing the recall to inferred relations that could not be extracted by Multi[2]OIE. The more inferred relations in the benchmark, the more beneficial merging extractions are.

We also evaluate our abstractive OpenIE models

| OpenIE Model | Documents | MRR | P@1 | Hit@5 |
|---|---|---|---|---|
| Multi[2]OIE | Top 10 | 0.193 | 0.127 | 0.267 |
| Abstractive OIE4 | Top 10 | 0.154 | 0.080 | 0.240 |
| Abstractive Back Translated OIE4 | Top 10 | 0.167 | **0.100** | 0.227 |
| Abstractive SuRE Augmented OIE4 | Top 10 | 0.157 | 0.093 | 0.220 |
| Abstractive SuRE Augmented Back Translated OIE4 | Top 10 | **0.181** | 0.093 | **0.287** |

Table 7: Performance of QUEST on the CQ-W dataset using the Top 10 Google documents (Lu et al., 2019).

on only the inferred relations within each benchmark. To do this, we remove non-inferred relations from the gold standards. We can only measure the resulting recall of the models because the models are trained to generate both inferred and non-inferred relations and the metrics we use penalize the precision when there are too many predicted relations for a given sentence, which would be the case for any sentence that had non-inferred relations. Figure 3 shows the results of these experiments. As before, the more inferred relations in the benchmark, the better suited an abstractive OpenIE model is for the task.

Upon a manual examination of the generated relations of each model, we observe that fine-tuning T5 on SuRE-augmented data results in generated relations replacing some of its predicates with the predicates from SuRE. Table 8 demonstrates one example of a model generating a predicate that does not exist within the sentence but is a common predicate among the SuRE-augmented relations.

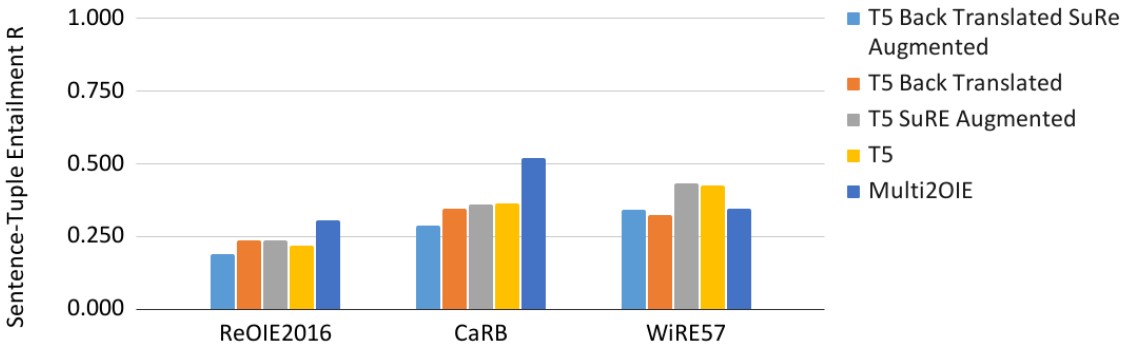

Figure 3: Comparison of Sentence-Tuple Entailment Recall of different combinations of OpenIE models on only the inferred relations in the benchmarks. All models are trained on OIE4.

| Sentence | Formerly known as Edo, it has been the de facto seat of government since 1603 when Shogun Tokugawa Ieyasu made the city his headquarters. |
| --- | --- |
| **T5 Fine-Tuned on OIE4** | (it; has been; the de facto seat of government since 1603) |
| **T5 Fine-Tuned on SuRE-Augmented OIE4** | (it; is also known as; the de facto seat of government since 1603) |

Table 8: A demonstration that T5 fine-tuned on OIE4 augmented with SuRE extractions generates predicates from the SuRE extractions rather than the sentence. This sentence is from the WiRE57 test set.

### Case Study

To further test the applicability of abstractive OpenIE, we evaluate its performance on QUEST, a downstream Complex QA task that uses OpenIE in its pipeline (Lu et al., 2019). QUEST specifically desires higher recall from its OpenIE model, which can be achieved by extracting inferred relations. We show the results in Table 7. The results show that augmenting the training data improves downstream performance, indicating that including more inferred relations in the training data is helpful for this task.

## 6   Conclusion

In this paper, we introduce abstractive OpenIE, an alternative to what we call extractive OpenIE, the paradigm all current OpenIE models currently follow, in order to address the problems of inferred relations and surface form extraction. We find that existing OpenIE datasets and metrics are ill-suited for this task. As a result, we introduce abstractive training set, model, and metrics. We then compare our models trained on different abstractive training sets and the state-of-the-art extractive OpenIE model using preexisting OpenIE benchmarks. Overall, we find that our models achieve higher performance on inferred relations, which extractive OpenIE models have previously struggled with. We believe abstractive OpenIE has potential as a task that will greatly benefit downstream applications that use OpenIE in their pipeline.

## 7   Limitations

In this work, we used a relatively smaller T5-base model. A model with more parameters may have led to improved performance. Further, the corpora we chose are all limited to English. As a result, our results are not generalizable to any downstream task that relies on different languages.

### Ethics Statement

We did not create any of the models, datasets, or applications covered in this paper. Any ethical issues with the preexisting OpenIE datasets we use in this paper will reflect on this work.

### Acknowledgements

This material is based upon work supported by the National Science Foundation IIS 16-19302 and IIS 16-33755, Zhejiang University ZJU Research 083650, Futurewei Technologies HF2017060011 and 094013, IBM-Illinois Center for Cognitive Computing Systems Research (C3SR) and IBM-Illinois Discovery Accelerator Institute (IIDAI), grants from eBay and Microsoft Azure, UIUC OVCR CCIL Planning Grant 434S34, UIUC CSBS Small Grant 434C8U, and UIUC New Frontiers Initiative. Any opinions, findings, conclusions, or recommendations expressed in this publication are

those of the author(s) and do not necessarily reflect the views of the funding agencies.

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

## A Empirical Results

We present our empirical results in tables 9, 10, and 11.

| Model | Training Set | Benchmark | CaRB Score | | | Sentence-Tuple Entailment | | | Combined Tuple-Tuple Entailment | | | Tuple-Tuple Entailment | | |
|---|---|---|---|---|---|---|---|---|---|---|---|---|---|---|
| | | | P | R | F1 | P | R | F1 | P | R | F1 | P | R | F1 |
| Multi$^2$OIE | OIE4 | LSOIE-wiki | 0.396 | 0.318 | 0.353 | 0.953 | 0.381 | 0.545 | 0.595 | 0.488 | 0.536 | 0.591 | 0.467 | 0.522 |
| Abstractive T5 | OIE4 | LSOIE-wiki | 0.496 | 0.369 | 0.423 | 0.964 | 0.432 | 0.596 | 0.614 | 0.525 | 0.566 | 0.608 | 0.499 | 0.548 |
| Abstractive T5 | OIE4 Back Translated | LSOIE-wiki | 0.5 | 0.483 | 0.491 | 0.961 | 0.439 | 0.603 | 0.627 | 0.546 | 0.584 | 0.640 | 0.510 | 0.568 |
| Abstractive T5 | OIE4 with SuRE Relations | LSOIE-wiki | 0.518 | 0.49 | 0.504 | 0.963 | 0.436 | 0.601 | 0.632 | 0.565 | 0.597 | 0.645 | 0.511 | 0.570 |
| Abstractive T5 | OIE4 Back Translated with SuRE Relations | LSOIE-wiki | 0.538 | 0.527 | 0.532 | 0.974 | 0.571 | 0.720 | 0.645 | 0.670 | 0.657 | 0.660 | 0.611 | 0.634 |
| Multi$^2$OIE | OIE4 | ReOIE2016 | 0.565 | 0.373 | 0.449 | 0.939 | 0.351 | 0.511 | 0.835 | 0.504 | 0.629 | 0.763 | 0.477 | 0.587 |
| Abstractive T5 | OIE4 | ReOIE2016 | 0.733 | 0.449 | 0.557 | 0.953 | 0.425 | 0.588 | 0.861 | 0.580 | 0.693 | 0.779 | 0.543 | 0.640 |
| Abstractive T5 | OIE4 Back Translated | ReOIE2016 | 0.706 | 0.565 | 0.628 | 0.948 | 0.418 | 0.580 | 0.855 | 0.582 | 0.693 | 0.806 | 0.531 | 0.640 |
| Abstractive T5 | OIE4 with SuRE Relations | ReOIE2016 | 0.757 | 0.572 | 0.652 | 0.953 | 0.424 | 0.587 | 0.871 | 0.602 | 0.712 | 0.814 | 0.531 | 0.643 |
| Abstractive T5 | OIE4 Back Translated with SuRE Relations | ReOIE2016 | 0.813 | 0.647 | 0.72 | 0.976 | 0.574 | 0.723 | 0.894 | 0.736 | 0.808 | 0.823 | 0.684 | 0.747 |
| Multi$^2$OIE | OIE4 | CaRB | 0.525 | 0.309 | 0.389 | 0.935 | 0.357 | 0.517 | 0.856 | 0.538 | 0.661 | 0.682 | 0.487 | 0.568 |
| Abstractive T5 | OIE4 | CaRB | 0.619 | 0.336 | 0.436 | 0.949 | 0.431 | 0.593 | 0.882 | 0.592 | 0.709 | 0.694 | 0.526 | 0.599 |
| Abstractive T5 | OIE4 Back Translated | CaRB | 0.592 | 0.394 | 0.473 | 0.945 | 0.422 | 0.583 | 0.843 | 0.578 | 0.686 | 0.682 | 0.491 | 0.571 |
| Abstractive T5 | OIE4 with SuRE Relations | CaRB | 0.619 | 0.389 | 0.478 | 0.951 | 0.428 | 0.591 | 0.862 | 0.584 | 0.697 | 0.701 | 0.495 | 0.580 |
| Abstractive T5 | OIE4 Back Translated with SuRE Relations | CaRB | 0.647 | 0.442 | 0.525 | 0.975 | 0.572 | 0.721 | 0.884 | 0.707 | 0.786 | 0.702 | 0.619 | 0.658 |
| Multi$^2$OIE | OIE4 | WiRe57 | 0.45 | 0.343 | 0.389 | 0.960 | 0.362 | 0.526 | 0.668 | 0.572 | 0.617 | 0.378 | 0.574 | 0.456 |
| Abstractive T5 | OIE4 | WiRe57 | 0.519 | 0.357 | 0.423 | 0.988 | 0.355 | 0.523 | 0.665 | 0.613 | 0.638 | 0.361 | 0.586 | 0.447 |
| Abstractive T5 | OIE4 Back Translated | WiRe57 | 0.502 | 0.399 | 0.445 | 0.946 | 0.475 | 0.632 | 0.642 | 0.675 | 0.658 | 0.290 | 0.661 | 0.403 |
| Abstractive T5 | OIE4 with SuRE Relations | WiRe57 | 0.506 | 0.391 | 0.441 | 0.981 | 0.469 | 0.635 | 0.633 | 0.670 | 0.651 | 0.284 | 0.678 | 0.401 |
| Abstractive T5 | OIE4 Back Translated with SuRE Relations | WiRe57 | 0.537 | 0.37 | 0.439 | 0.990 | 0.371 | 0.539 | 0.665 | 0.611 | 0.637 | 0.377 | 0.556 | 0.449 |

Table 9: Empirical results of different models on different benchmarks. Differences in the number of inferred relations in each of the benchmarks influences the relative performance of each model. The benchmarks are listed from lowest to highest proportion of relations with inferred predicates.

| Model | Training Set | Benchmark | CaRB Score | | | Sentence-Tuple Entailment | | | Combined Tuple-Tuple Entailment | | | Tuple-Tuple Entailment | | |
|---|---|---|---|---|---|---|---|---|---|---|---|---|---|---|
| | | | P | R | F1 | P | R | F1 | P | R | F1 | P | R | F1 |
| Multi$^2$OIE | OIE4 | Re-OIE2016 | 0.813 | 0.647 | 0.72 | 0.976 | 0.574 | 0.723 | 0.894 | 0.736 | 0.808 | 0.823 | 0.684 | 0.747 |
| Multi$^2$OIE + Abstractive T5 | OIE4 | Re-OIE2016 | 0.601 | 0.561 | 0.581 | 0.972 | 0.563 | 0.713 | 0.869 | 0.737 | 0.798 | 0.791 | 0.675 | 0.729 |
| Multi$^2$OIE + Abstractive T5 | OIE4 Back Translated | Re-OIE2016 | 0.64 | 0.554 | 0.594 | 0.964 | 0.678 | 0.796 | 0.864 | 0.789 | 0.825 | 0.757 | 0.722 | 0.739 |
| Multi$^2$OIE + Abstractive T5 | OIE4 with SuRE Relations | Re-OIE2016 | 0.584 | 0.569 | 0.577 | 0.962 | 0.591 | 0.732 | 0.860 | 0.756 | 0.805 | 0.788 | 0.689 | 0.735 |
| Multi$^2$OIE + Abstractive T5 | OIE4 Back Translated with SuRE Relations | Re-OIE2016 | 0.599 | 0.542 | 0.569 | 0.965 | 0.683 | 0.800 | 0.846 | 0.768 | 0.805 | 0.764 | 0.701 | 0.731 |
| Multi$^2$OIE | OIE4 | CaRB | 0.647 | 0.442 | 0.525 | 0.975 | 0.572 | 0.721 | 0.884 | 0.707 | 0.786 | 0.702 | 0.619 | 0.658 |
| Multi$^2$OIE + Abstractive T5 | OIE4 | CaRB | 0.525 | 0.404 | 0.457 | 0.970 | 0.568 | 0.717 | 0.859 | 0.699 | 0.770 | 0.678 | 0.621 | 0.648 |
| Multi$^2$OIE + Abstractive T5 | OIE4 Back Translated | CaRB | 0.544 | 0.402 | 0.463 | 0.958 | 0.684 | 0.798 | 0.874 | 0.767 | 0.817 | 0.651 | 0.670 | 0.661 |
| Multi$^2$OIE + Abstractive T5 | OIE4 with SuRE Relations | CaRB | 0.518 | 0.413 | 0.46 | 0.961 | 0.595 | 0.735 | 0.851 | 0.719 | 0.780 | 0.670 | 0.630 | 0.649 |
| Multi$^2$OIE + Abstractive T5 | OIE4 Back Translated with SuRE Relations | CaRB | 0.531 | 0.409 | 0.462 | 0.957 | 0.690 | 0.802 | 0.859 | 0.757 | 0.805 | 0.634 | 0.663 | 0.648 |
| Multi$^2$OIE | OIE4 | WiRe57 | 0.537 | 0.37 | 0.439 | 0.990 | 0.371 | 0.539 | 0.665 | 0.611 | 0.637 | 0.377 | 0.556 | 0.449 |
| Multi$^2$OIE + Abstractive T5 | OIE4 | WiRe57 | 0.481 | 0.376 | 0.422 | 0.992 | 0.621 | 0.764 | 0.625 | 0.755 | 0.684 | 0.264 | 0.732 | 0.388 |
| Multi$^2$OIE + Abstractive T5 | OIE4 Back Translated | WiRe57 | 0.476 | 0.39 | 0.429 | 0.988 | 0.564 | 0.718 | 0.639 | 0.730 | 0.681 | 0.312 | 0.691 | 0.430 |
| Multi$^2$OIE + Abstractive T5 | OIE4 with SuRE Relations | WiRe57 | 0.482 | 0.39 | 0.431 | 0.958 | 0.600 | 0.738 | 0.651 | 0.728 | 0.688 | 0.283 | 0.722 | 0.407 |
| Multi$^2$OIE + Abstractive T5 | OIE4 Back Translated with SuRE Relations | WiRe57 | 0.458 | 0.398 | 0.426 | 0.948 | 0.656 | 0.775 | 0.633 | 0.727 | 0.677 | 0.294 | 0.723 | 0.418 |

Table 10: Empirical results where the relations extracted by Multi$^2$OIE and abstractive OpenIE are combined. Redundant relations are removed after the combination of extractions. Redundant relations are relations that are entailed by at least one other relation in the same sentence. If two relations entail each other, the shorter one is removed.

| Model | Training Set | Benchmark | CaRB Score | Sentence-Tuple Entailment | Combined Tuple-Tuple Entailment | Tuple-Tuple Entailment |
|---|---|---|---|---|---|---|
| | | | R | R | R | R |
| Multi²OIE | OIE4 | ReOIE2016 Inferred Predicates or Args | 0.231 | 0.304 | 0.452 | 0.411 |
| Abstractive T5 | OIE4 | ReOIE2016 Inferred Predicates or Args | 0.231 | 0.219 | 0.394 | 0.343 |
| Abstractive T5 | OIE4 Back Translated | ReOIE2016 Inferred Predicates or Args | 0.152 | 0.236 | 0.424 | 0.380 |
| Abstractive T5 | OIE4 with SuRE Relations | ReOIE2016 Inferred Predicates or Args | 0.223 | 0.236 | 0.386 | 0.408 |
| Abstractive T5 | OIE4 Back Translated with SuRE Relations | ReOIE2016 Inferred Predicates or Args | 0.087 | 0.189 | 0.481 | 0.475 |
| Multi²OIE | OIE4 | CaRB Inferred Predicates or Args | 0.116 | 0.520 | 0.641 | 0.605 |
| Abstractive T5 | OIE4 | CaRB Inferred Predicates or Args | 0.109 | 0.362 | 0.522 | 0.482 |
| Abstractive T5 | OIE4 Back Translated | CaRB Inferred Predicates or Args | 0.082 | 0.346 | 0.509 | 0.495 |
| Abstractive T5 | OIE4 with SuRE Relations | CaRB Inferred Predicates or Args | 0.128 | 0.360 | 0.511 | 0.490 |
| Abstractive T5 | OIE4 Back Translated with SuRE Relations | CaRB Inferred Predicates or Args | 0.099 | 0.289 | 0.532 | 0.498 |
| Multi²OIE | OIE4 | WiRe57 Inferred Predicates or Args | 0.051 | 0.346 | 0.546 | 0.536 |
| Abstractive T5 | OIE4 | WiRe57 Inferred Predicates or Args | 0.059 | 0.427 | 0.544 | 0.599 |
| Abstractive T5 | OIE4 Back Translated | WiRe57 Inferred Predicates or Args | 0.057 | 0.325 | 0.501 | 0.523 |
| Abstractive T5 | OIE4 with SuRE Relations | WiRe57 Inferred Predicates or Args | 0.067 | 0.433 | 0.574 | 0.605 |
| Abstractive T5 | OIE4 Back Translated with SuRE Relations | WiRe57 Inferred Predicates or Args | 0.043 | 0.341 | 0.468 | 0.509 |

Table 11: Empirical results of models where the gold standard consists only of relations with inferred predicates or arguments. We only measure recall in this case because relations are extracted per-sentence, so relations that do not have inferred predicates will also be extracted, which will lower the precision.