# OpenReview forum: "Abstractive Open Information Extraction"
_EMNLP/2023/Conference — EMNLP 2023 Main_

### Official Review · Reviewer_TQSQ · 2023-08-05

**Soundness:** 3

**Excitement:**

3: Ambivalent: It has merits (e.g., it reports state-of-the-art results, the idea is nice), but there are key weaknesses (e.g., it describes incremental work), and it can significantly benefit from another round of revision. However, I won't object to accepting it if my co-reviewers champion it.

**Missing References:**

[1] Del Corro, L., & Gemulla, R. (2013, May). Clausie: clause-based open information extraction. In Proceedings of the 22nd international conference on World Wide Web (pp. 355-366).
[2]  Xavier, C. C., Strube de Lima, V. L., & Souza, M. (2015). Open information extraction based on lexical semantics. Journal of the Brazilian Computer Society, 21, 1-14.
[3] Sena, C. F. L., & Claro, D. B. (2020). PragmaticOIE: a pragmatic open information extraction for Portuguese language. Knowledge and Information Systems, 62(9), 3811-3836.
questions
[4] Kolluru, K., Mohammed, M., Mittal, S., & Chakrabarti, S. (2022, May). Alignment-augmented consistent translation for multilingual open information extraction. In Proceedings of the 60th Annual Meeting of the Association for Computational Linguistics (Volume 1: Long Papers) (pp. 2502-2517).

**Paper Topic And Main Contributions:**

The present paper proposes a dataset, a model, and evaluation metrics for Abstractive Open Information Extraction, or as more common in the literature OIE with synthetic relations. The authors create a dataset for OIE by data augmentation off the OIE dataset through back-translation of sentences and extraction of new tuples employing a system for traditional relation extraction trained on the original corpus. Further, the authors propose and implement a model for abstractive OIE by fine-tunning the T5 model on the proposed corpus in a simialr fashion to what is done by Kolluru et al in GEN2OIE. additionally the authors propose two semantic-based evaluation metrics employing BERT-based textual entailment model and compare their proposal with Multi2OIE on different datasets using both CaRB lexical metric and their proposed three metrics.

**Reasons To Accept:**

I think the focus of extracting synthetic relations is interesting and few work have investigated the topic, especially in the neural approaches, but it is not clear that the proposed data augmentation is actually helping with this since it removes implicit and non-lexicalised relations in the entailment-based denoising. The improvement in performance on CaRB, for example, may be due to more examples allowing the model to overcome ordering problem of relations for the model
Regarding the evaluation metrics, I completely agree with the authors on the necessity of semantic-based metrics for the area. I suggest the authors to think how to combine their idea with structure preservation of the extractions to guarantee semantic coherence between the extraction and the gold standard

TL;DR
Focus on the extraction of synthetic relations in OIE, which is often neglected in the literature;
Proposes a semantics-based evaluation metric.

**Reasons To Reject:**

First, it is important to point out that I find the main argument of the authors regarding the limitation of the task and the available models to stem mostly from unknowing the relevant literature. However, I agree with them that there is a heavy focus on "extractive" OIE, especially recently due to OIE as a labelling approach that became popular with available datasets for OIE. Examples of work on OIE that deal with synthetic relations are present in the literature since earlier approaches, e.g. [1,2], or newer approaches, e.g. [3,4], both for the English language and others. Second, while the ability to extract implicit or non-lexicalised relations is an important feature for OIE extractors and datasets, the argument that current manually curated datasets have too few such relations and the proposal to artificially inflate their number doesn't seem natural or even reasonable. The authors' proposal for the generation of new triples through back-translation paraphrasing relies on the fact that no new information is included in the dataset. Such a resource only makes sense in an "extractive" paradigm, as the main problem for "abstractive" models would thus be how to evaluate their extractions, in my point of view - a problem the authors accurately identify. Notice that a small number of extraction per text/sentence is *not* a problem faced by OIE. As such, there is little incentive to artificially create more extractions. As such, I find the justification of the work flawed.

Regarding the generation of the dataset, it is not clear why the authors employ translation models, which are more prone to errors than simply paraphrase technology. Also, by relying on extractions on traditional relation extraction technology in a context of a high diversity of relations with similar semantics, there seems to be a high introduction of noise in the dataset. Further, the authors fail to discuss the training and testing procedures of the SuRE model or report on its performance.

Regarding the evaluation metrics, the proposed metrics have similar shortcomings to previous ones, such as that of Stanovsky and Dagan or of CaRB. OIE is a structural task, meaning that the results are structured. By concatenating the parts of the tuple, i.e. conflating the structure, in order to evaluate entailment, we lose the structure. As such, extractions with different semantic content such as (Salleri, is, a city in Nepal) -this informs on the category of things or concept that the individual Salleri participates-  and (Salleri, is a city in, Nepal) -this informs a located_in relationship between the entities Salleri and Nepal- are treated as the same.

In Section 5, pg 7 col 2, the authors seem to manipulate the vertical alignment of the text block as well as avoid using subsections in order to fit their context within the 8-page limit, making the text in the first line of each paragraph clash with the preceding paragraph.

The authors only consider the English language in their work. English is relatively poor in terms of syntax, and for a task that is so dependent on syntax, such as OIE, the performance of the method may stem from the language characteristics. Also, since the authors are limited to the English language, in my opinion, they should very explicitly state that, as their results may simply not generalise to OIE as a whole but to OIE in English.

TL;DR
The motivation is flawed, and the authors do not seem to have a good grasp of the related literature;
The creation of the dataset is not clearly discussed, and the noise control performed by the authors seems to make their data augmentation processing innocuous as any implicit and non-lexicalised relation, i.e. the ones they are trying to capture, would be cancelled, thus only introducing paraphrases of existing relations in the dataset;
The evaluation metrics are, in my opinion, ill-suited for the task;
The authors only consider the English language, which is a considerable limitation for such a syntax-reliant task as OIE and the authors do not take into consideration this limitation in their work.

**Reproducibility:**

3: Could reproduce the results with some difficulty. The settings of parameters are underspecified or subjectively determined; the training/evaluation data are not widely available.

**Reviewer Confidence:**

5: Positive that my evaluation is correct. I read the paper very carefully and I am very familiar with related work.

**Typos Grammar Style And Presentation Improvements:**

- references to tables and figures in the text should be capitalised.
-pg 3 col 2 line 204: Closed Relation Extraction -> this is not a common term to the area. Either use Traditional Relation Extraction or simply Relation Extraction
pg 7 line 497: T he -> The
- section 5: get rid of the negative \vspaces
How are the prompts for the segmented narratives constructed? in which sense do they differ from the nin-segmented narratives?
What does Figure 3 means? What is measured on X-axis in the graphs? What do the authors mean by partial observability and full observability?

---

> ### Author Rebuttal · Authors · 2023-08-29
>
> First, we would like to thank the reviewer for your constructive criticism and feedback. However, we respectfully disagree with the statements that the "Motivation is flawed" or "Justification of the work is flawed", "evaluation metrics are ill-suited for the task", and "authors do not take into consideration this limitation [English]".
>
> We believe there are some misunderstandings with regards to the data generation process we describe in this paper. We are not using back translation to generate new triples, but rather to generate new sentences that maintain the same relations as the sentences they are derived from. We will rewrite this section to be clearer in the camera ready version.
>
> We do not inflate the number of relations in the benchmarks. Instead, we are attempting to rectify deficiencies in existing OpenIE models in extracting certain relations by introducing inferred relations and transforming existing extractive relations in training sets in a scalable way.
>
> Regarding your first point, we agree that there is previous work with inferred relations. However, of those approaches, ClausIE, LSOE, and PragmaticOIE are rule-based methods which have since been replaced in favor of supervised OpenIE models. We consider them similar to OpenIE6, a supervised OpenIE model which has baked-in components to deal with specific inferred relations containing [is], [of], and [from]. Our aim is to introduce a general method for extracting inferred relations that does not require manually pre-specifying exceptions so that it is readily usable to different domains. We will cite these as relevant previous works in our camera-ready paper. We agree that Gen2OIE is a very relevant model that is able to extract inferred relations, and we will cite it and include it in our experiments in our camera-ready paper.
>
> Regarding your second point, the creation of relation tuples is strictly for the training sets, not the benchmarks we are evaluating on. All OpenIE training sets have been weakly labeled. As a result, there is no guarantee that the gold standards for a given sentence in an OpenIE training set constitute all relations within that sentence. This can be observed by the fact that SuRE, a relation extraction model that can only extract a predefined set of relations, is still able to extract new relations from sentences in the training set that are not entailed by any existing gold standard relations in that sentence. Table 2 shows that this data augmentation increases the number of relations in the training set by roughly 10\%, which is not a trivial amount.
>
> Regarding generation of the dataset, the purpose of back translation is not to generate new relations, but rather a way to generate a new sentence where the old relation tuples could become inferred. The sentence retains the same meaning and the relation tuples stay the same so that we can be reasonably confident that even if the relation tuples contain tokens no longer in the back translated sentence, they still can be extracted from that sentence. The additional validation steps are intended to protect against noise resulting from the back translation process. It is by design that the meaning of the relation tuples and the sentence are kept the same. We use an off-the-shelf SuRE model and state in lines 488-489 that we chose it because it is a state-of-the-art relation extraction model. We will clarify that it is off-the-shelf within the camera-ready version.
>
> Regarding why we chose back translation, we required that the paraphrasing method would retain proper nouns while changing the syntax and word choice of the sentence. We previously experimented with different sentence simplification methods like MUSS, but found that they either did not retain the proper nouns or did not substantially change the syntax of the sentence. Back translation is another paraphrasing method that has been previously studied which we decided to use because it suited our particular needs.
>
> Regarding the evaluation metrics, we agree that the parts of the tuple are best evaluated part-to-part rather than combining them before measuring entailment. However, entailment models behave very unpredictably when the hypothesis and premise are very short. For example, using our entailment model "is" does not entail "is" very much, giving around a 0.5 entailment score. Similarly, "Nepal" and "Nepal" gives around a 0.55 entailment score. Given that many predicates in all datasets are single tokens and some arguments are single tokens, this makes it very difficult to obtain accurate scores unless you combine the parts of each tuple to form phrases. Our purpose is to introduce a new task, and the creation of a new metric that can perform this part-to-part comparison would be an interesting direction for future research.
>
> Regarding the limitation to English, we agree that our work is not generalizable to other languages, as we mention in the Limitations section, lines 585-588.
>
> We also thank the reviewer for pointing out style issues and typos. We will fix these in the camera-ready version.

---

### Official Review · Reviewer_GwEy · 2023-08-05

**Soundness:** 4

**Excitement:**

3: Ambivalent: It has merits (e.g., it reports state-of-the-art results, the idea is nice), but there are key weaknesses (e.g., it describes incremental work), and it can significantly benefit from another round of revision. However, I won't object to accepting it if my co-reviewers champion it.

**Paper Topic And Main Contributions:**

In this paper, the authors proposed and developed an abstractive OpenIE (Open Information Extraction) model, in contrast to traditional ones, which are extractive. First, a dataset specifically for abstractive IE is built via two methods, back translation and data augmentation via relation extraction (RE). They make use of generative models, e.g. T5, to generate relations from a sentences. First, predicates are extracted from the sentence. Then, for each predicate the model predicts the arguments for the relation with that predicate. Results showed that the proposed abstractive OpenIE models achieved higher performance on inferred relations, which previous extractive OpenIE struggled with.

The paper is well-written and explains clearly the related works, motivation, as well as the proposed approach.

A thorough analysis is needed. It should show that the proposed model does not harm the performance of explicit, or un-inferred relations.
The approach is interesting and will probably be useful for the community. However, the proposed method relies on quite many other tasks, which may lead to propagated errors and not adaptable to other domains and languages.

**Questions For The Authors:**

Results showed that the proposed abstractive OpenIE models achieved higher performance on inferred relations, which previous extractive OpenIE struggled with.
The paper is well-written and explains clearly the related works, motivation, as well as the proposed approach.
A thorough analysis is needed. It should show that the proposed model does not harm the performance of explicit, or un-inferred relations.
The approach is interesting and will probably be useful for the community. However, the proposed method relies on quite many other tasks, which may lead to propagated errors and not adaptable to other domains and languages.

**Reasons To Accept:**

Results showed that the proposed abstractive OpenIE models achieved higher performance on inferred relations, which previous extractive OpenIE struggled with.
The paper is well-written and explains clearly the related works, motivation, as well as the proposed approach.

**Reasons To Reject:**

A thorough analysis is needed. It should show that the proposed model does not harm the performance of explicit, or un-inferred relations.
The approach is interesting and will probably be useful for the community. However, the proposed method relies on quite many other tasks, which may lead to propagated errors and not adaptable to other domains and languages.

**Reproducibility:**

3: Could reproduce the results with some difficulty. The settings of parameters are underspecified or subjectively determined; the training/evaluation data are not widely available.

**Reviewer Confidence:**

4: Quite sure. I tried to check the important points carefully. It's unlikely, though conceivable, that I missed something that should affect my ratings.

---

> ### Author Rebuttal · Authors · 2023-08-29
>
> First, we would like to thank the reviewer for your constructive criticism and feedback. We feel encouraged that you find our contributions will be useful to the NLP community. All concerns raised will be fully addressed in the camera-ready version.
>
> Q: A thorough analysis is needed. It should show that the proposed model does not harm the performance of explicit, or un-inferred relations.
>
> Thank you for this suggestion. We agree that looking at the performance on uninferred and inferred relations only would be an interesting topic to explore further. We have thought of these experiments, but the task definition makes performing these experiments tricky. Because OpenIE works on a per-sentence level, we have previously investigated performance on sentences with at least one inferred relation. Our observations were that the performance on sentences with at least one inferred relation was highly correlated with the performance on all sentences, so we did not include those results.  For instance, Multi2OIE outperformed all Abstractive OpenIE models in sentence-tuple entailment F1 score for each benchmark except WiRE57, where T5 trained on SuRE-augmented OIE4 had 0.617 and Multi2OIE had 0.524.
>
> We also performed an experiment where we computed recall for only the inferred relations in the benchmarks, and found that T5 trained on SuRE-augmented OIE4 had the highest sentence-tuple entailment recall for WiRE57, with 0.433 compared to 0.346 for Multi2OIE. On CaRB, T5 trained on SuRE-augmented OIE4 had 0.350 recall and Multi2OIE had 0.519, and on ReOIE2016 T5 trained on SuRE-augmented OIE4 had 0.238 recall and Multi2OIE had 0.306. With the addition of extra space in the camera-ready paper, we will add these experiments.
>
> As for non-inferred relations, we have not investigated whether performance on non-inferred relations is degraded by Abstractive OpenIE. With the addition of extra space in the camera-ready paper, we will add these experiments.
>
>
>
> Q: The proposed method relies on quite many other tasks, which may lead to propagated errors and not adaptable to other domains and languages.
>
> We would like to draw attention to the fact that all OpenIE training sets are generated using weak labeling. We agree that adding more components to the pipeline will lead to more errors. However, manually labeling from scratch is a very tedious task and we show that though noisy there is a way to construct Abstractive OpenIE training dataset at scale. We also attempted to minimize noise by testing the entailment of the back translation and SuRE augmentation we used when constructing an abstractive training dataset. For the evaluation metric, we also rely on entailment. However, evaluating LLM generations is still an active area of research. Once a better evaluation metric is available, we will gladly use it.
>
> As mentioned in the limitation section, this approach may not be suitable for other languages, but future works may still take inspiration from this one.

---

### Official Review · Reviewer_8EaF · 2023-08-05

**Soundness:** 4

**Excitement:**

3: Ambivalent: It has merits (e.g., it reports state-of-the-art results, the idea is nice), but there are key weaknesses (e.g., it describes incremental work), and it can significantly benefit from another round of revision. However, I won't object to accepting it if my co-reviewers champion it.

**Paper Topic And Main Contributions:**

The paper has two main contributions. First, they introduce an abstractive Open Information Extraction (OpenIE) task as opposed to the extractive open information task, which is a comparatively easier problem. Second, they extend the existing dataset to create a new abstractive OpenIE dataset and a semantic-based evaluation metric to train and evaluate models on them for extracting abstractive relations.
• The paper is well-motivated, and the problem statement is clear.
• They describe the dataset creation task in detail.
• Experiments support the authors' claims. They show that combining the results of the trained model with the augmented abstractive dataset and previous extractive models improves the final prediction on inferred relations. They also study the impact of each augmentation approach separately on benchmark test datasets.


**Questions For The Authors:**

• Line 292-294 is not clear, as the gold standard already indicates that there is an entailment relationship with the original sentence. Please clarify why you add this extra computation.
• It would have been nicer to conduct more experiments in downstream tasks to showcase the trained model on the created abstractive OIE dataset.
• In the experiments in Figure 1, the T5 trained model with "backtranslated and SuRe augmented" underperforms even the model trained with the original data. If this happens, please explain what the benefit is of creating datasets using both approaches for this task?
• It would be nice to evaluate the trained model on a proportion of the datasets that only contain inferred relations to better understand the benefits of different approaches for augmentation (before combining the models, more analysis on inferred relations is required).
• Why does Multi2OIE outperform the trained models on 0.271(CaRB)? Isn't it expected that on this dataset, the T5 trained model on abstractive data gets closer to or better than Multi2OIE? Qualitative analysis can help understand the model performances.
• It would be nice to add another benchmark test dataset, which is subsampled from the augmented abstractive OIE4 dataset, and evaluate the trained models on the same dataset given they are from the same distribution.
• Adding qualitative analysis is useful to understand the limitations of the model.


**Reasons To Accept:**

• New methods for evaluating models in the abstractive OpenIE task are explored, which is encouraging for future research in this area.
• The approach for augmenting data and the proposed evaluation metrics are significant contributions and could be useful in any area within the community.
• The dataset is evaluated on a downstream question-answering task, which demonstrates the different applications of the dataset.
• The quality of the augmented data is comprehensively evaluated and checked.
• In general, the authors present clear contributions, and the paper is well-motivated. It is nicely structured, and the problem and proposed solution in each section are clear and easy to understand.


**Reasons To Reject:**

The limited scope of the experiments is not sufficient to assess if the method would work equally well in other scenarios (models and downstream tasks). Additionally, there is no qualitative analysis to help understand the limitations and strengths of the annotation task, dataset, and the trained model.

**Reproducibility:**

3: Could reproduce the results with some difficulty. The settings of parameters are underspecified or subjectively determined; the training/evaluation data are not widely available.

**Reviewer Confidence:**

4: Quite sure. I tried to check the important points carefully. It's unlikely, though conceivable, that I missed something that should affect my ratings.

**Typos Grammar Style And Presentation Improvements:**

In Section 3.1, you have not mentioned the dataset (OIE4) you used to create your training data, although you mentioned it in the tables.

---

> ### Author Rebuttal · Authors · 2023-08-29
>
> We sincerely thank your review and feedback on our paper. We feel encouraged that you find our contributions significant to the NLP community. All concerns raised will be fully addressed in the camera-ready version.
>
> Question: Line 292-294 is not clear, as the gold standard already indicates that there is an entailment relationship with the original sentence. Please clarify why you add this extra computation.
>
> Response 1: Thank you for catching this typo. It is meant to be whether the paraphrased sentence entails the gold standard relations. This is intended to ensure the backtranslated sentence contains the same relations as the original sentence. We will fix this in the camera-ready version.
>
> Question: It would have been nicer to conduct more experiments in downstream tasks to showcase the trained model on the created abstractive OIE dataset.
>
> Response 2: We agree it would be nicer to add more experiments on downstream tasks. We wanted to use an OpenIE application that was recent and had a quantitative metric for evaluation. Unfortunately, when we reached out to different authors for their code for several applications that use OpenIE we were only able to get code for QUEST. When we gain access to further applications we will analyze the impact of the created Abstractive OpenIE dataset.
>
> Question:  In the experiments in Figure 1, the T5 trained model with "backtranslated and SuRe augmented" underperforms even the model trained with the original data. If this happens, please explain what the benefit is of creating datasets using both approaches for this task?
>
> Response 3: This is a very good point. Figure 1 is more like an ablation study to understand whether backtranslation, SuRE-augmentation or both together results in a high quality Abstractive OpenIE training dataset. All these derived datasets are useful, representing different levels of abstraction. We anticipate that a better abstractive model (better than T5) can benefit from any/all of these derived datasets.
>
> Question: It would be nice to evaluate the trained model on a proportion of the datasets that only contain inferred relations to better understand the benefits of different approaches for augmentation (before combining the models, more analysis on inferred relations is required).
>
> Response 4: Thank you for this suggestion. We have thought of these experiments, but the task definition makes performing these experiments tricky. Because OpenIE works on a per-sentence level, we have previously investigated performance on sentences with at least one inferred relation. Our observations were that the performance on sentences with at least one inferred relation was highly correlated with the performance on all sentences, so we did not include those results.  For instance, Multi2OIE outperformed all Abstractive OpenIE models in sentence-tuple entailment F1 score for each benchmark except WiRE57, where T5 trained on SuRE-augmented OIE4 had 0.617 and Multi2OIE had 0.524.
>
> We also performed an experiment where we computed recall for only the inferred relations in the benchmarks, and found that T5 trained on SuRE-augmented OIE4 had the highest sentence-tuple entailment recall for WiRE57, with 0.433 compared to 0.346 for Multi2OIE. On CaRB, T5 trained on SuRE-augmented OIE4 had 0.350 recall and Multi2OIE had 0.519, and on ReOIE2016 T5 trained on SuRE-augmented OIE4 had 0.238 recall and Multi2OIE had 0.306. With the addition of extra space in the camera-ready paper, we will add these experiments.
>
> Question: Why does Multi2OIE outperform the trained models on 0.271(CaRB)? Isn't it expected that on this dataset, the T5 trained model on abstractive data gets closer to or better than Multi2OIE? Qualitative analysis can help understand the model performances.
>
> Response 5: The performance of Multi2OIE is better than Abstractive OpenIE on every benchmark except 0.338 (WiRE57), which contains the highest number of inferred relations among test sets. Upon manual inspection, we find that the annotation of relations among different test sets is not consistent, which is one of the reasons why the performance of all OpenIE models on WiRE57 is lower than on other benchmarks. We will include this qualitative analysis in the camera-ready version.
>
> Our focus is on inferred relations in the datasets
> We want models to be good on datasets with no inferred relations, but for that we might need better higher-parameter models.
> This is a first step towards abstractive OpenIE
> Expect better models in the future
>
> Question: It would be nice to add another benchmark test dataset, which is subsampled from the augmented abstractive OIE4 dataset, and evaluate the trained models on the same dataset given they are from the same distribution.
>
> Response 6: Thank you for this suggestion. The benchmarks we use are all manually annotated or annotated using crowd sourcing, while the training data are all weakly labeled. As a result, we believe the gold standards for the benchmarks are more likely to be correct than the gold standards for the training data, which do sometimes contain labeling errors such as typos or ignoring negations like not or never. As a result, we do not want to introduce a new gold standard based on potentially erroneous training data and instead focus on evaluating on existing benchmarks because they already contain some inferred relations and we can be more certain of their correctness.
>
> Question: Adding qualitative analysis is useful to understand the limitations of the model.
>
> Response 7: Thank you for this suggestion. We did perform some and found that the abstractive model tends to copy the training data, such as using relations from SuRE when fine-tuned on SuRE augmented data. We will include qualitative analysis in the camera-ready version.
>
> We also thank the reviewer for pointing out style issues and typos. We will fix these in the camera-ready version.

---

### Meta-Review · Area_Chair_n7Bm · 2023-09-20

**Recommendation:** 4

**Metareview:**

This paper focuses on abstractive OIE where the relation may not explicitly appear in the text. The research is meaningful and interesting. Most of reviewers express their concerns about the experiments. The experiments are not sufficient and need further improvements according to the reviewers' comments.

---

### Decision · Program_Chairs · 2023-10-07

**Decision:**

Accept-Main

**Comment:**

This paper focuses on abstractive OIE where the relation may not explicitly appear in the text. The research is meaningful and interesting. Most of reviewers express their concerns about the experiments. The experiments are not sufficient and need further improvements according to the reviewers' comments.